**Data Availability Statement:** This manuscript's minimal data set is not publicly available to ensure patient confidentiality. The data underlying the

# The Temporary Incapacity (TI) register as a complementary system to traditional epidemiological surveillance during the COVID-19 pandemic in Spain

**Dante Culqui Lévano** ⬤ *, **Sofía Escalona López, Alín Gherasim, Jesús Oliva Domínguez, María Teresa Disdier Rico, Montserrat García Gómez**

Subdirectorate General for Environmental Health and Occupational Health, Directorate General for Public Health, Ministry of Health, Madrid, Spain

* dculqui@gmail.com, danteroger@hotmail.com

## Abstract

### Introduction

During the COVID-19 pandemic, a set of social measures were adopted for the preservation of business activity and the protection of workers. One of these measures was issuing the Temporary Disability (TD) for COVID-19 cases, close contacts, and especially vulnerable workers.

### Objetive

This study analyzes whether the TD registry could be used as a complementary source to traditional epidemiological surveillance.

### Methods

A longitudinal study of time series was carried out with a cross-correlation analysis of TD and COVID-19 cases reported to the National Epidemiological Surveillance Network (RENAVE). The analysis included six pandemic waves between 10/03/2020 and 31/12/2021 in Spain. Cross-correlation coefficients (r) were calculated using a time lag of -14 days.

### Results

During the study period, 2,253,573 TD processes were recorded in Spain and 4,894,802 COVID-19 cases were reported to RENAVE. Significant positive correlations were observed at time lags of -7, -10, and -14, indicating that TD notification preceded RENAVE notification. In the first and sixth pandemic waves, TD notification preceded RENAVE by 12 and 7 days, respectively. Negative correlations between the two series were observed in the second and fourth waves, coinciding with a lower number of reported cases. In the third and fifth waves, TD notification also preceded RENAVE (lags -1, -5 and -14, -7, respectively).

results presented in the study are available from National Social Security Institute on request to authors who meet the standards and complete a request form. Requests for data can be placed at the following URL: https://sede.seg-social.gob.es/wps/portal/sede/sede/Ciudadanos/2022estadisticas/090316_c_ai?changeLanguage=en.

**Funding:** The author(s) received no specific funding for this work.

**Competing interests:** The authors have declared that no competing interests exist.

## Conclusions

The results confirm the usefulness of TD registry as a complementary system to traditional epidemiological surveillance in Spain, by detecting COVID-19 cases in the 7, 10, and 14 days prior. A better positive correlation is observed in waves where more cases were reported.

## Introduction

The COVID-19 was declared by WHO as Public Health Emergency of International Concern (PHEIC) on 30 January 2020 and further characterized as a pandemic on 11 March 2020 [1]. Due to the rapid spread of this disease both nationally and internationally, the member states of the European Union quickly adopted coordinated emergency measures to protect public health and prevent the collapse of the economy.

In Spain, the government authorities decreed the general confinement of the population along with the almost total closure of economic activities in an attempt to slow down the transmission of the virus. Efforts were also made to improve the disease detection and reporting. During February and early March 2020, coordinated prevention and control measures were strengthened within the framework of the Interterritorial Council of the National Health System (CISNS) of Spain in order to improve the sensitivity of case detection, limit public presence at certain events, and recommend travel restrictions, among other measures [2].

In parallel to the state of emergency, it was necessary to activate a set of social measures to preserve businesses and protect workers. One of these measures was the social protection of workers who had to isolate due to COVID-19, to being a close contact of a case of the disease, or an especially sensitive (vulnerable) worker. It took the form of Temporary Disability (TD) assimilated to occupational accident sick leave [3,4].

On March 10, 2020, the government issued a royal decree-law stating that the situation of people forced to stay at home for health reasons would be considered as a TD due to occupational accident [5]. In addition, on March 12, another royal decree-law included urgent measures to reinforce the health system, support families and directly affected companies. Resources worth more than 18,000 million euros were mobilized, including a reinforcement of the health sector of approximately 3,800 million euros and measures to provide cash and reduce costs for companies, especially small and medium-sized, self-employed workers, and the tourism sector [6]. Likewise, the Administration´s capacity to react to extraordinary situations was reinforced, streamlining the procedure for the adquisition of any type of goods and services that were necessary [6].

The Social Security system processed 5.6 million TDs related to COVID-19 from March 10, 2020 to December 31, 2021. Of these, 2.2 million were due to infection, and 3.4 million were due to close contacts with confirmed COVID-19 cases [7].

When studying TD, it should be noted that these sick leaves are granted for infection, close contact with a confirmed case of COVID-19, or for being an especially sensitive worker, and their analysis goes beyond economic studies, as it covers areas such as psychology, general health, and social behavior [8].

During the pandemic, the main source of information on COVID-19 morbidity was the National Epidemiological Surveillance Network (RENAVE). However, it was observed that the first case reported through RENAVE was recorded eight days after the first TD was granted for infection. In addition, from the beginning of the pandemic until April 21, 2020, the number of TDs due to COVID-19 reported was higher than the number of cases included in RENAVE [9].

They are two independent systems with different objectives. RENAVE depends on the health authorities and seeks epidemiological surveillance. TD system depends on the social security authorities and guarantees the economic income of workers on sick leave from the first day.

There is evidence of the usefulness of complementary sources to traditional epidemiological surveillance systems. Systems such as influenza sentinel doctors, or the notification system called "medical hotline," a phone-based case reporting system, have been used as complementary systems to the traditional surveillance system [10]. Some authors have assessed the usefulness of monitoring work absenteeism as an early warning tool for flu outbreaks [11,12]. They also argue that it is an important tool for providing information about the socioeconomic impact of the COVID-19 pandemic [11] and a complex indicator of the well-being of the working-age population [13]. Other authors have proposed studying the antecedents of previous sick leave as predictors of long-term sick leave [14]. In the United States, some experts have recommended linking various sources of information with traditional epidemiological surveillance, as this could maximize case detection and deepen understanding of disease characteristics [15].

This cross-correlation study between the time series of TD registry and RENAVE, has the aim of identifying whether the TD registry could be-used as a complementary information source for traditional epidemiological surveillance, in order to improve the capacity for early detection of cases and epidemic outbreaks in the future. This study represents a preliminary analysis that we intend to continue to complete as the pandemic progresses.

## Methodology

We performed a longitudinal time series study. We used the daily incidence rate of COVID-19 in patients between 16 and 65 years old in Spain (the age group that benefits from TD, in order to allow the comparison of indicators) as the dependent variable, referred to as IRRenave. It includes the following ICD-10 diagnoses (ICD-10-ES codes): Infection due to coronavirus, unspecified (B34.2), Coronavirus associated with SARS as the cause of diseases classified under another concept (B97.21), and COVID-19 2020 (U07.1) [16]. The daily IRRenave was calculated per 100,000 inhabitants in Spain, based on the number of cases registered in RENAVE, divided by the total population on the same day and in the same age group (16 to 65 years old).

As the independent variable, we used the daily incidence rate of TD, referred to as IRTD, which corresponded to workers between 16 and 65 years diagnosed with COVID-19, excluding the TDs due to quarantine and particularly sensitive workers. The daily IRTD was calculated for 100,000 social security affiliates.

The first TD due to COVID-19 in Spain was registered on February 15, 2020, followed by several days without any recorded TD, nor COVID-19 notified cases in RENAVE. For this reason, and to compare daily rates, our study period ranged from March 10, 2020, to December 31, 2021 (662 days).

We used two sources of information: the first, the RENAVE series of COVID-19 cases in people aged 16 to 65 years provided by the National Center of Epidemiology, Institute of Health Carlos III. We used the population between 16 and 65 years old, available on the National Statistics Institute (INE) website, as denominator for the rate calculation [17]. The second information source providing the registered TD, was supplied in electronic format by the National Social Security Institute (INSS). The figures of the monthly affiliated population to social security [7] were used as denominator for the rate calculation in this case.

We performed a descriptive analysis of both series to identify whether the mean and variability remain constant over time. The sequence graph of both series and the autocorrelation

analysis were performed. Both IRRenave and IRTD series were transformed; initial tests were applied with one difference to eliminate trends and identify stationarity. In addition, we used the "expert modeler" function of SPSS (to explore ARIMA models), in order to identify any model that could be stationary in the mean. Once the variables were transformed into stationary series, the cross-correlation analysis was performed between the residuals of the models obtained from the two variables IRTD_1 (with one transformation) and IRRenave_2 (with two transformations). The transformations are performed in order to identify the stationarity of the series, in order to identify a series with white noise. The transformation can be set with one lag, 2 lags (Lag), or as many lags as necessary, until a time series is found that could be useful for cross-correlation analysis. Finally, we chose the best analysis model in terms of biological coherence. From the analysis of the complete series (from 10.03.20 to 31.12.21), partial analyses were carried out for each of the six pandemic wavesand for the pre-vaccination and post-vaccination periods, delimiting the following periods:

  Complete series: From 10.03.20 to 31.12.21.
  First epidemic wave: From 10.03.20 to 26.06.20.
  Second epidemic wave: From 27.06.20 to 06.12.20.
  Third epidemic wave: From 07.12.20 to 07.03.21.
  Fourth epidemic wave: From 08.03.21 to 20.06.21.
  Fifth epidemic wave: From 21.06.21 to 09.10.21.
  Sixth epidemic wave: From 10.10.21 to 31.12.21.
  Pre-vaccination period: From 10.03.20 to 26.12.20.
  Post-vaccination period: From 27.12.20 to 31.12.21.

  Cross-correlation "r" coefficients were calculated [18,19] (that inform about the degree and positive or negative direction of the association between time series and range from +1 to -1) using a time lag of up to 14 days. The incubation period for the SARS-CoV-2 virusgenerally ranges from 2 to 14 days. This means that, after being exposed to the virus, most individuals will develop symptoms within that timeframe. That's why an incubation period of up to 14 days is used to encompass the full range in which some individuals may develop symptoms.

  Graphs showing the lagged correlations were obtained. The level of statistical significance is indicated by horizontal lines parallel to the X axis. We used SPSS 27.0 and STATA 16.0 for time series analysis; the tables were constructed using Excel 16.0.

  The authors did not have access to information that could identify individual participants during or after data collection.

## Results

Between March 10, 2020, and December 31, 2021, a total of 2,253,573 COVID-19-related TDs were registered in Spain, and 4,894,802 COVID-19 cases were reported to RENAVE. The TD rate in this period was 11,909.56 /100,000 affiliates, and the RENAVE rate was 25,867.78 COVID-19 cases/100,000 inhabitants between 16 and 65 years old.

  Fig 1 shows the description of the complete series of 638 days, in which the six waves studied, the confinement period (March 15, 2020, to June 21, 2020), as well as the pre-vaccination period (March 10, 2020, to December 26, 2021), post-vaccination period (December 27, 2020, to December 31, 2021), and the emergence of the Omicron variant (first case: November 27, 2021) are observed.

  Fig 2 shows the sequence plots and simple autocorrelation plots of both series. Two transformations were performed on the IRRenave series, and the variable was renamed IRRenave_2, while one transformation was applied to the IRTD series, and the variable was renamed IRTD_1. After the transformations, stationarity in the mean of both series was observed.

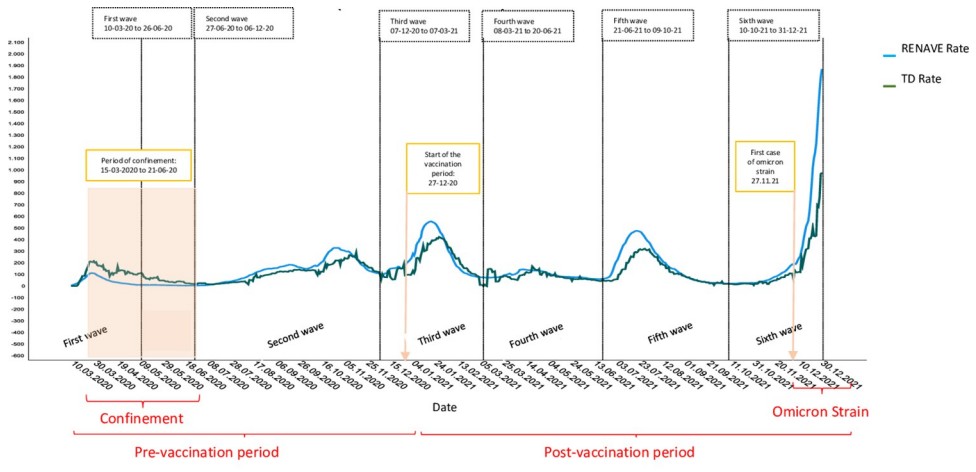

**Fig 1. Incidence rate of COVID-19-related TDs and COVID-19 cases notified to RENAVE.** Spain, 10.03.2020 to 31.12.2021.

The residuals (white noise) extracted from the daily incidence rates of COVID-19 throughout Spain were studied.

In the cross-correlation analysis of the entire series, we found significant positive correlation at lag zero (0). This means that the TDs are a good indicator of the COVID-19 cases reported to RENAVE throughout the period. Additionally, significant positive correlations were observed at lags -7, -10, and -14 (Fig 3A and Table 1), which could indicate that the notification of the TDs preceded the corresponding notification in RENAVE.

a.- Sequency graph for IRRenave_2 serie (with two transformations).

b.- Sequency graph IRTD_1 (with one transformation).

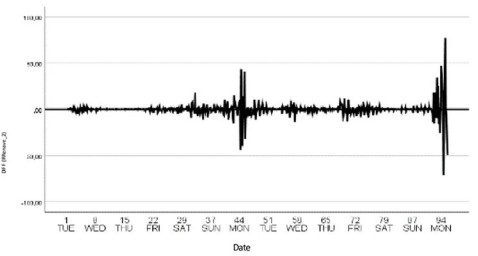

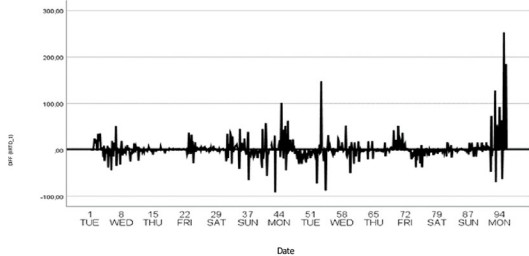

c.- Simple autocorrelation graph IRRenave_2.

d.- Simple autocorrelation graph IRTD_1

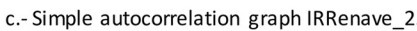

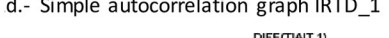

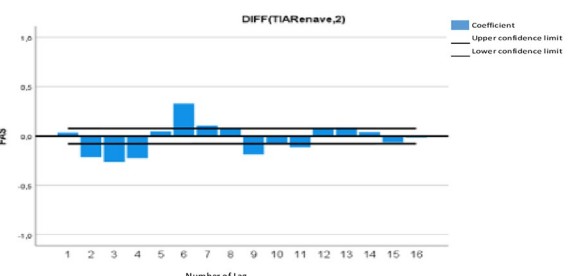

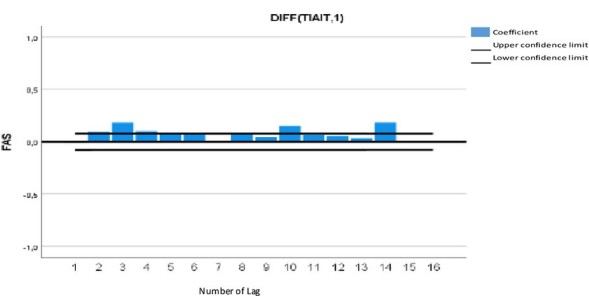

**Fig 2. Descriptive analysis of time series: incidence rates of cases in RENAVE (IRRenave_2) and incidence rates of Temporary Disability (IRTD_1).** Spain, 10.03.2020 to 31.12.2021.

When performing the correlation analysis by waves, positive cross-correlation was observed in the first and sixth waves. In the first wave, the notification of TDs anticipated the notification of RENAVE by up to 12 days (lags -12 and -6), and in the sixth wave, by up to one week (lag -7) (Fig 3B and 3G).

In the third wave, it was observed that the notification of TDs also preceded RENAVE at lags -1 and -5, and in the fifth wave, up to two weeks in advance (lags -14, -7. However, in both waves, negative cross-correlation was also observed on day -2 (Fig 3D and 3F and Table 1). Negative cross-correlations between the two series were observed in the second and fourth waves, with lags -14, -10, and -2 for the second wave, and lag -7 for the fourth wave (Fig 3C and 3E and Table 1).

The pre- and post-vaccination periods were also studied. However, the results of the analysis do not appear to have an influence on the notifications of the TD or COVID-19 cases (Table 2).

## Discussion

Our results show a significant cross-correlation between TD due to COVID-19 infection in workers and the COVID-19 cases notified to RENAVE. The significant positive correlation at lag 0, is particularly noteworthy; it means that TDs are a good indicator of COVID-19 cases reported to RENAVE when considering the entire study period. Additionally, depending on the wave noted, significant positive correlations were observed in the previous 7, 10, and 14 days, indicating that the notification of TD anticipated the epidemiological surveillance conducted through RENAVE, up to two weeks in advance.

The epidemiological surveillance system in Spain was overwhelmed in the first waves of the pandemic. There were not enough PCR tests and the case definitions was very strictly defined, reducing the number of people tested nearly only to those with severe disease admitted to hospitals as well as Health Care Workers. It´s highly probable that there was a sizable amount of underestimation of COVID-19 cases.

Furthermore, Spain institutes a special situation for business called "ERTE", a temporary regulatory employment file, where all or a significant number of non-essential workers are entitled to stay at home because of a temporal closure of the economic activity due to the lockdown. It is likely that many of these workers do not request for a TD leave, albeit they could have been infected and/or close contact of a COVID-19 cases. Both system: being under an ERTE and the TD are incompatible.

In a previous study carried out in healthcare workers in Spain [9], it was observed that TD due to COVID-19 in the first weeks, captured better the epidemic intensity, than RENAVE, as TDs due to infection were higher in number and rate than the cases recorded in RENAVE since the beginning of the pandemic. The granting of TD by clinical and epidemiological investigation of suspected cases, often by telephone, without diagnostic confirmation, may explain why the number of TDs was higher and earlier in time than that registered in RENAVE, where only confirmed cases were recorded. This meant that, especially in the first wave (with a shortage of diagnostic tests), but also in subsequent waves, more cases were captured and the incidence increased earlier by the TD registry than in RENAVE.

It should also be remembered that the results of the first round of the national seroprevalence study ENE-COVID [20] estimated the sensitivity of the surveillance system during the first pandemic wave at 9.7% (95% CI: 8.96% - 10.29%) [18]. That is, the real number of infections estimated at the peak of the first wave would be close to 100,000 per day, compared to the 10,000 reported to the system. The milder cases, probably younger people who did not require medical attention, were not recorded by RENAVE, however, if being essential workers, they had to record the TD to justify work absence.

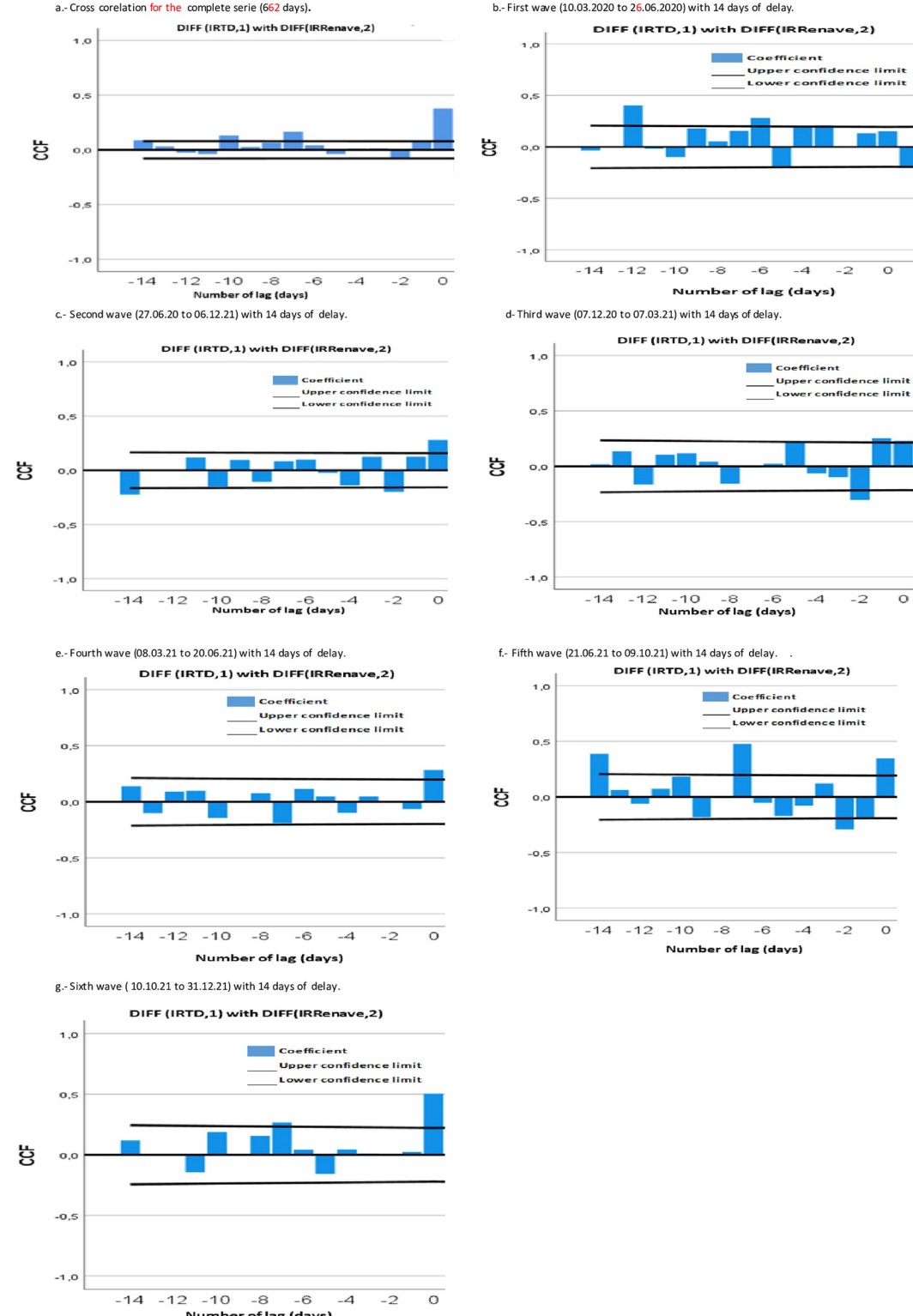

**Fig 3. Cross-correlation analysis of the complete time series and by pandemic waves.** Spain, 10.03.2020 to 31.12.2021.

**Table 1. Summary table of cross-correlation analysis of the complete series and by waves during the COVID-19 pandemic.** Spain, March 10, 2020, to December 31, 2021.

| Wave analysed | Days of lag with positive cross-correlation | Days of lag with negative cross-correlation |
|---|---|---|
| Complete period of study: | (-14,-10,-7) | |
| First wave | (-12, -6) | |
| Second wave | | (-14,-10, -2) |
| Third wave | (-5,-1) | (-2) |
| Fourth wave | | (-7) |
| Fifth wave | (-14,-7) | (-2) |
| Sixth wave | (-7) | |

In this context, the analysis of TD could complement traditional surveillance, since often the symptoms of a disease and not the detection of the specific virus, determine the start of a TD. Studies carried out in France [21], Belgium [22], and the United Kingdom (UK) [23] show the usefulness of analyzing TD in the detection of influenza outbreaks.

In France, 92% of the reported outbreaks were detected, with an average of 2.5 weeks earlier than traditional surveillance. Even monitoring absences from work due to influenza-like illness and/or general discomfort, allowed the detection of flu outbreaks up to 5 weeks before the detection of cases through traditional surveillance. In recent years, the usefulness of complementary systems to traditional surveillance for various diseases and risk factors has been confirmed. For example, syndromic surveillance was used to identify the impact of heat waves and pollution on the population seeking medical attention in the UK [12].

In another study in Miami-United States of America (USA), information on work absenteeism was applied as a non-specific syndromic indicator of the occurrence of influenza and other infectious diseases in the community [24].

A relevant finding of our study is that the time delay identified in the correlation of the series coincides with the incubation period of the disease, established at an average of six days from symptom onset, with an infectious peak occurring two days before symptom onset [25], considering that at the beginning of the pandemic it spanned even more days. In the case of the analyzed series, during the first wave, the delay that best correlated both series occurred on days -12 and -6, and in the sixth wave, the delay occurred on day -7.

The global pattern described showed a different distribution in the six waves studied. It was observed that during the 1st, 3rd, 5th, and 6th waves, the notified TD anticipated the traditional RENAVE surveillance system, with a positive sign. These results confirm information from studies conducted in Spain, indicating an earlier reporting of COVID-19 related TD than the information collected by the traditional surveillance system (RENAVE) [9]. In other periods, as both the improvements in the traditional surveillance system and the pandemic situation changed, RENAVE recorded higher incidence rates than the TD system.

Negative correlation was observed on days -14, -10, and -2 in the second wave, and on day -7 in the 4th wave. The second wave had a smaller peak (lower number of cases) compared to the third wave, and the fourth wave had the lowest peak of all the waves studied. The lower number of cases in the second and fourth waves compared to the other waves could explain why the reporting of cases to RENAVE did not collapse, as it could have happened in the other periods. Regarding this, in comparative evaluations of the traditional influenza surveillance system with the TD registry, a strong correlation was observed, which was more evident at the peaks of the studied curves, even though the case frequencies were significantly different [11]. That is, the model fits better when TD reporting is higher and is even more useful in detecting outbreaks [11,12,26,27], results that coincide with our findings, in which a better positive

**Table 2. Cross-correlation analysis, complete series and by waves with positive and negative signs.** Spain, 10.03.20 to 31.12.21.

Cross correlation for serie: DIFF (IRTD,1) with DIFF(IRRenave,2)

Waves with correlated days and trends

| Lag | All pandemic process — Total serie (638 days) — Cross correlation Total serie | Standard error | First wave cross correlation | Standard error | Second wave cross correlation | Standard error | Third wave cross correlation | Standard error | Fourth wave cross correlation | Standard error | Fifth wave cross correlation | Standard error | Sixth wave cross correlation | Standard error | Post-vaccination — Pre-vaccination cross correlation | Standard error | Post-vaccination — Post-vaccination cross correlation | Standard error |
|---|---|---|---|---|---|---|---|---|---|---|---|---|---|---|---|---|---|---|
| -14 | **0,087** (*) | 0,039 | -0,036 | 0,104 | **-0,224** (-) | 0,082 | 0,019 | 0,117 | 0,140 | 0,106 | **0,387** (*) | 0,103 | 0,118 | 0,122 | -0,105 | 0,060 | 0,134 | 0,053 |
| -13 | 0,032 | 0,039 | 0,006 | 0,103 | -0,004 | 0,082 | 0,136 | 0,116 | -0,101 | 0,105 | 0,062 | 0,102 | -0,006 | 0,121 | 0,026 | 0,060 | 0,006 | 0,053 |
| -12 | -0,027 | 0,039 | **0,404** (*) | 0,103 | 0,005 | 0,082 | -0,165 | 0,115 | 0,090 | 0,105 | -0,061 | 0,102 | -0,003 | 0,120 | 0,055 | 0,060 | -0,050 | 0,053 |
| -11 | -0,038 | 0,039 | -0,015 | 0,102 | 0,118 | 0,082 | 0,105 | 0,115 | 0,100 | 0,104 | 0,073 | 0,101 | -0,145 | 0,120 | 0,034 | 0,060 | -0,045 | 0,053 |
| -10 | **0,130** (*) | 0,039 | -0,099 | 0,102 | **-0,150** (-) | 0,081 | 0,117 | 0,114 | -0,143 | 0,104 | 0,182 | 0,101 | 0,187 | 0,119 | -0,087 | 0,060 | 0,153 | 0,053 |
| -9 | 0,029 | 0,039 | 0,178 | 0,101 | 0,096 | 0,081 | 0,043 | 0,113 | -0,007 | 0,103 | -0,183 | 0,100 | 0,005 | 0,118 | 0,094 | 0,060 | 0,019 | 0,053 |
| -8 | 0,066 | 0,039 | 0,051 | 0,101 | -0,107 | 0,081 | -0,156 | 0,113 | 0,077 | 0,103 | -0,011 | 0,100 | 0,155 | 0,117 | -0,016 | 0,060 | 0,073 | 0,053 |
| -7 | **0,165** (*) | 0,039 | 0,156 | 0,100 | 0,084 | 0,081 | -0,010 | 0,112 | **-0,191** (-) | 0,102 | **0,479** (*) | 0,099 | **0,269** (+) | 0,116 | -0,023 | 0,060 | 0,189 | 0,053 |
| -6 | 0,042 | 0,039 | **0,282** (*) | 0,100 | 0,099 | 0,080 | 0,022 | 0,111 | 0,116 | 0,102 | -0,051 | 0,099 | 0,042 | 0,115 | 0,002 | 0,060 | 0,051 | 0,053 |
| -5 | -0,037 | 0,039 | -0,202 | 0,099 | -0,024 | 0,080 | **0,221** (*) | 0,110 | 0,046 | 0,101 | -0,170 | 0,098 | -0,158 | 0,115 | 0,014 | 0,059 | -0,054 | 0,052 |
| -4 | 0,013 | 0,039 | 0,194 | 0,099 | -0,139 | 0,080 | -0,064 | 0,110 | -0,099 | 0,101 | -0,077 | 0,098 | 0,045 | 0,114 | 0,015 | 0,059 | 0,001 | 0,052 |
| -3 | 0,015 | 0,039 | 0,208 | 0,098 | 0,125 | 0,080 | -0,100 | 0,109 | 0,047 | 0,100 | 0,122 | 0,097 | 0,010 | 0,113 | 0,095 | 0,059 | 0,023 | 0,052 |
| -2 | -0,079 | 0,039 | 0,007 | 0,098 | **-0,199** (-) | 0,079 | -0,304 | 0,108 | -0,004 | 0,100 | -0,290 | 0,097 | -0,002 | 0,113 | -0,173 | 0,059 | -0,072 | 0,052 |
| -1 | 0,077 | 0,039 | 0,134 | 0,097 | 0,125 | 0,079 | 0,254 (*) | 0,108 | -0,063 | 0,099 | -0,187 | 0,096 | 0,023 | 0,112 | 0,022 | 0,059 | 0,088 | 0,052 |
| 0 | 0,378 | 0,039 | 0,152 | 0,097 | 0,280 | 0,079 | 0,232 | 0,107 | 0,284 | 0,099 | 0,347 | 0,096 | 0,505 | 0,111 | 0,319 | 0,059 | 0,390 | 0,052 |

(*) Correlation with significant positive trend

(-) Correlation with significant negative trend

correlation was observed in waves where there are more studied patients (1st, 3rd, 5th, and 6th waves).

Our study has some limitations that should be considered when analyzing the results: it does not include information by gender, information on unemployed people, or other groups of interest or risk, such as the elderly and children, because they cannot benefit from TD. We did not take into account any comorbidities either. Despite the evident usefulness of TD as a complementary tool to the traditional surveillance system, some authors consider that TD only reflects part of the course of epidemics or outbreaks by analyzing the working-age population groups of a country and not all age or risk groups that could be affected by a disease [11]. However, despite its limitations, in countries where TD has been analyzed as a complementary system to the traditional system, the analysis results have proven to be reliable and useful [11,24,28].

Another limitation is the relatively short period of time that we studied so the information on TD does not pertain to the Ministry of Health. We are aware that the analysis would be more robust if we had observations for a longer period of time. The obtained cross-correlation values must be analyzed carefully, especially in individual wave analyses, where the series are short. Additionally, there may be variables not analyzed in this study that could influence the observed results. Predictive model studies using TD have shown that such models predict much better when a larger dataset is used, in terms of the number of observations and covariates [28]. Despite this limitation, we consider that our results justify the routine use of TD as a complementary information system to the surveillance traditional. Having real-time access to this data could strengthen infectious disease surveillance.

We can conclude that, despite the limitations of this study, the results of this research confirm the usefulness of TD as a system that complements traditional epidemiological surveillance in Spain (RENAVE) and had the virtue of detecting COVID-19 cases in advance. The use of information from the TD registry to monitor epidemic curves and as a system that alerts the beginning of an increase in infection incidence could be considered as an early warning system, useful for epidemiological and public health surveillance.

## Supporting information

**S1 File.**
(DOCX)

## Acknowledgments

We would like to thank the National Institute of Social Security for providing us with data on TD. We would also like to thank the National Center for Epidemiology and the Coordinating Center for Health Alerts and Emergencies for providing us with corresponding data on cases included in the National Epidemiological Surveillance Network.

## Author Contributions

**Conceptualization:** Dante Culqui Lévano, Montserrat García Gómez.

**Data curation:** Dante Culqui Lévano, Jesús Oliva Domínguez, Montserrat García Gómez.

**Formal analysis:** Dante Culqui Lévano, Sofía Escalona López, Alín Gherasim, Montserrat García Gómez.

**Funding acquisition:** Dante Culqui Lévano.

**Investigation:** Dante Culqui Lévano, Sofía Escalona López, Alín Gherasim, Montserrat García Gómez.

**Methodology:** Dante Culqui Lévano, Sofía Escalona López, Alín Gherasim, Montserrat García Gómez.

**Project administration:** Sofía Escalona López, Alín Gherasim, Montserrat García Gómez.

**Resources:** Montserrat García Gómez.

**Software:** Montserrat García Gómez.

**Supervision:** Dante Culqui Lévano, Montserrat García Gómez.

**Validation:** Dante Culqui Lévano, Sofía Escalona López, Jesús Oliva Domínguez, María Teresa Disdier Rico, Montserrat García Gómez.

**Visualization:** Jesús Oliva Domínguez, María Teresa Disdier Rico, Montserrat García Gómez.

**Writing – original draft:** Dante Culqui Lévano.

**Writing – review & editing:** Dante Culqui Lévano, Sofía Escalona López, Alín Gherasim, Jesús Oliva Domínguez, María Teresa Disdier Rico, Montserrat García Gómez.

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
