## [Decision Letter · Decision Letter 0]

23 Jan 2024

PONE-D-23-27681Temporary Disability (TD) registry as a complementary system to traditional epidemiological surveillance during the COVID-19 pandemic in Spain a National Study.PLOS ONE

Dear Dr. Culqui L.,

Thank you for submitting your manuscript to PLOS ONE. After careful consideration, we feel that it has merit but does not fully meet PLOS ONE’s publication criteria as it currently stands. Therefore, we invite you to submit a revised version of the manuscript that addresses the points raised during the review process.

The manuscript has merit and generally exhibits a well-structured format. It addresses pertinent issues that could enhance our understanding of the post-COVID landscape. However, the three ad hoc reviewers have pointed out the need for improvements in the text, particularly regarding its comparability with other research in the field. They also highlight the necessity for greater clarity in elucidating concepts, methods, and limitations of the study, aspects with which I fully agree upon my review. Therefore, I suggest that the authors carefully consider the reviewers' suggestions and implement them to refine the presentation of the manuscript.

We look forward to receiving your revised manuscript.

Kind regards,

Ricardo de Mattos Russo Rafael, Ph.D.

Academic Editor

PLOS ONE

2. Thank you for stating the following in your Competing Interests section: "NO authors have competing interests".

3. Please amend your manuscript to include your abstract after the title page.

Reviewers' comments:

Reviewer's Responses to Questions

**Comments to the Author**

1. Is the manuscript technically sound, and do the data support the conclusions?

Reviewer #1: Yes

Reviewer #2: Partly

Reviewer #3: Yes

2. Has the statistical analysis been performed appropriately and rigorously? 

Reviewer #1: Yes

Reviewer #2: I Don't Know

Reviewer #3: Yes

3. Have the authors made all data underlying the findings in their manuscript fully available?

Reviewer #1: Yes

Reviewer #2: Yes

Reviewer #3: Yes

4. Is the manuscript presented in an intelligible fashion and written in standard English?

Reviewer #1: Yes

Reviewer #2: Yes

Reviewer #3: Yes

5. Review Comments to the Author

Reviewer #1: INTRODUCTION

1. In the following excerpt: “The Social Security system processed 4.18 million TDs related to COVID-19 from January 2020 to May 1, 2021. Of these, 1.4 million were due to the infection and 2.78 million were due to close contacts with confirmed COVID -19 cases [7]”, why does the data presented have the cutoff point of May 1, 2021 and not the end of the period observed by the study? It would make more sense to present the updated data until December 31, 2021.

METHODOLOGY

2. In the excerpt: “…We used the daily incidence rate of COVID-19 in patients between 16 and 65 years old in Spain as the dependent variable, referred to as IRRenave”, complete with the information “per 100,000 inhabitants”, as also in the excerpt: “As the independent variable, we used the daily incidence rate of TD, referred to as IRTD…”. The information that the indicator was calculated per 100,000 inhabitants only appears in the fifth paragraph, which doesn't make much sense.

3. It is necessary to explain that in the IRRenave indicator, the inclusion criteria for the age group - from 16 to 65 years old – is due to the fact that it is the age group that benefits from TD, in order to allow the comparison of indicators. This is not clear in the Methodology and is only explained in the Discussion.

4. I suggest moving the section “The daily IRRenave was calculated per 100,000 inhabitants in Spain, based on the number of cases registered in RENAVE...”, in order to take into account what was exposed in item 2.

5. Correct “Both TDRenave and IRTD”, to “Both IRRenave and IRTD”.

RESULTS

6. In the paragraph: “Figure 1 shows the description of the complete series of 638 days…”, please review and correct the periods, which are defined differently from that exposed in the Methodology section. “…as well as the pre-vaccination period (March 10, 2020, to December 26, 2020), post-vaccination period (December 27, 2020, to December 31, 2021)”.

7. Correct the period of the first wave in the graph: “10-03-20 to 26-06-20”.

Reviewer #2: This is a short original article where a temporary disability registry (TD) is assessed as a complimentary system to the standard epidemiological surveillance during the first months of the COVID-19 pandemic in Spain.

The authors should try to improve some of the wording (i.e. Introduction, first paragraph: there is no declaration of a pandemic by WHO: WHO declares a PHEIC; COVID-19 is the disease: SARS-CoV-2 is the etiological agent and cause of the disease).

The main concerns related to this paper is the surveillance system that are compared as well as the statistical analysis.

1.- Surveillance system:

The epidemiological surveillance system in Spain was totally overwhelmed in the firsts waves of the pandemic. There were not enough PCR tests and the case definitions was very strictly defined, reducing the number of people tested nearly only to those with severe disease admitted to hospitals as well as Health Care Workers. Is is therefore widely admitted that there was a sizable amount of underestimation of COVID-19 cases.

The TD exceptional surveillance system relies in a personal declaration, usually by phone, where the individual with symptoms and/od that considers him o herself a close contact of a confirmed COVID-case asked for a Temporary Disability leave. There is no any clinical evaluation of the sick leave request, nor any confirmation of the likely diagnosis nor the status of close contact.

Furthermore, Spain institutes a special situation for business and other economic activities called “ERTE” a temporary regulatory employment file, where all or a significant number of workers are entitled to stay at home because of a temporal closure of the economic activity due to the lockdowns. It is likely that many of these workers do not request for a TD leave, albeit they could have been infected and/or close contact of a COVID-19 cases. Both system: being under an ERTE and the TD are incompatible.

In summary, neither the epidemiological surveillance system nor the TD complementary system were both reliable nor accurate systems for assessing the caseload of COVID-19 in Spain over the first waves of the pandemic. As an example, over the period March 10, 2020, to December 31, 2021, the authors state that 4,8 M COVID cases were reported to the epidemiological surveillance system. Our-World-In Data registry indicates more than 6 M COVID-19 cases over the same period.

Several of these limitations (and others as well) are discussed by the authors, but the main conclusion, i.e. that the TD registry could be a useful as a system to complement traditional epidemiological surveillance is weak enough if applied to the COVID-19 pandemic.

2.- Statistical analysis:

I suggest that the analysis should be reviewed by a statistician.

Reviewer #3: The manuscript aimed to explore whether Temporary Disability (TD) could complement the traditional Spanish surveillance system. The authors clearly defined the study's scope, connecting it to current literature. While the methodology and statistical analyses were generally appropriate, some adjustments are needed. More detailed presentation of results is warranted, and the discussion, though robust, brings the benefit from substantiation and comparisons with other studies, along with acknowledgment of significant limitations. This article underscores the potential of complementary systems, traditionally utilized in other countries, to serve as crucial early warning tools for epidemiological surveillance and public health in Spain.

Comments

1- Title suggestion: Temporary Disability (TD) registry as a complementary system to traditional epidemiological surveillance during the COVID-19 pandemic in Spain.

2- The keywords do not represent the entire manuscript content. Consider incorporating keywords such as Health Surveillance System or Epidemiological Monitoring for a more comprehensive representation.

3- Introduction (Page 3, 4th paragraph) - Insert reference in "In addition, on March 12, another royal decree-law included … and the tourism sector."

4- Introduction (Page 3, 7th paragraph) - It would be interesting to explain the delay and difference in the number of notifications between the systems. How do these systems work in Spain?

5- Methodology (Page 5, 2th paragraph) - Specify the age range used.

6- Methodology (Page 5, 6th paragraph) - Explain the transformations used in IRTD_1 e IRRenave_2.

7- Methodology (Page 6, 2th paragraph) - It would be interesting to explain better why the 14-day time interval and what was the level of significance adopted.

8- Results (Page 7, 2th paragraph) - Check the correct pre and post vaccination period.

9- Figure 1 - Insert the end of the observation period (31/12//2021) on the x-axis. Adjust the lines for the pre and post vaccination periods.

10- Figure 2 - It would be important to improve figure quality in order to better visualization.

11- Figure 3 - It would be important to improve the quality and shape of the figure. Furthermore, it's necessary to correct the 3 a) figure title. “Cross correlation for the complete series (638 days)”.

12- In Table 2, include a note and specify (*). Clarify any relevant information indicated by the asterisk.

6. PLOS authors have the option to publish the peer review history of their article (what does this mean?). If published, this will include your full peer review and any attached files.

Reviewer #1: No

Reviewer #2: No

Reviewer #3: No

---

## [Author Response · Author response to Decision Letter 0]

2 Feb 2024

Response to Reviewers

Dear Ms/Mr Editor,

We would like to take this opportunity to thank you for the time dedicated to our article.

We addressed all the comments and suggestions send by the reviewers, and we modified the article accordingly. Below, we are submitting the reviewer´s comments/suggestions and the performed changes:

1.- Journal requirements:

Comments to the Author 

Reply: We have reviewed the manuscript and we consider that it meets the PLOS ONE´s style requirements.

Comments to the Author 

2. Thank you for stating the following in your Competing Interests section: "NO authors have competing interests".

Reply: We have included the requested information in the cover letter and we thank you for making the modification on our behalf.

Comments to the Author 

3. Please amend your manuscript to include your abstract after the title page.

Reply: We included the abstract after the title page.

Reviewers' comments:

Reviewer's Responses to Questions

Comments to the Author

1. Is the manuscript technically sound, and do the data support the conclusions?

Reviewer #1: Yes

Reviewer #2: Partly

Reviewer #3: Yes

2. Has the statistical analysis been performed appropriately and rigorously?

Reviewer #1: Yes

Reviewer #2: I Don't Know

Reviewer #3: Yes

3. Have the authors made all data underlying the findings in their manuscript fully available?

Reviewer #1: Yes

Reviewer #2: Yes

Reviewer #3: Yes

4. Is the manuscript presented in an intelligible fashion and written in standard English?

Reviewer #1: Yes

Reviewer #2: Yes

Reviewer #3: Yes

5. Review Comments to the Author

Reviewer #1:

Comments to the Author

INTRODUCTION

1. In the following excerpt: “The Social Security system processed 4.18 million TDs related to COVID-19 from January 2020 to May 1, 2021. Of these, 1.4 million were due to the infection and 2.78 million were due to close contacts with confirmed COVID -19 cases [7]”.

Why does the data presented have the cutoff point of May 1, 2021 and not the end of the period observed by the study? It would make more sense to present the updated data until December 31, 2021.

Reply:

Thank you for pointing this out, it was a mistake. We have modified the text in order to correctly present Social Security data from 10th of March 2020 until December 31st, 2021.

Comments to the Author

METHODOLOGY

2. In the excerpt: “…We used the daily incidence rate of COVID-19 in patients between 16 and 65 years old in Spain as the dependent variable, referred to as IRRenave”, complete with the information “per 100,000 inhabitants”, as also in the excerpt: “As the independent variable, we used the daily incidence rate of TD, referred to as IRTD…”. The information that the indicator was calculated per 100,000 inhabitants only appears in the fifth paragraph, which doesn't make much sense.

Reply: 

Thank you very much for your comment. We have changed the order of the information for clarification as follows:

We used the daily incidence rate of COVID-19 in patients between 16 and 65 years old in Spain as the dependent variable, referred to as IRRenave. It includes the following ICD-10 diagnoses (ICD-10-ES codes): Infection due to coronavirus, unspecified (B34.2), Coronavirus associated with SARS as the cause of diseases classified under another concept (B97.21), and COVID-19 2020 (U07.1) [16]. The daily IRRenave was calculated per 100,000 inhabitants in Spain, based on the number of cases registered in RENAVE, divided by the total population on the same day and in the same age group (16 to 65 years old).

As the independent variable, we used the daily incidence rate of TD, referred to as IRTD, which corresponded to workers diagnosed with COVID-19, excluding the TDs due to quarantine and particularly sensitive workers. The daily IRTD was calculated for 100,000 social security affiliates.

3. It is necessary to explain that in the IRRenave indicator, the inclusion criteria for the age group - from 16 to 65 years old – is due to the fact that it is the age group that benefits from TD, in order to allow the comparison of indicators. This is not clear in the Methodology and is only explained in the Discussion.

Reply: 

Thank you for your comment. We included it in Methodology: 

We used the daily incidence rate of COVID-19 in patients between 16 and 65 years old in Spain (the age group that benefits from TD, in order to allow the comparison of indicators) as the dependent variable, referred to as IRRenave.

4. I suggest moving the section “The daily IRRenave was calculated per 100,000 inhabitants in Spain, based on the number of cases registered in RENAVE...”, in order to take into account what was exposed in item 2.

Reply: 

Done.

5. Correct “Both TDRenave and IRTD”, to “Both IRRenave and IRTD”.

Reply: 

Thank you, we corrected it in the manuscript.

Comments to the Author

RESULTS

6. In the paragraph: “Figure 1 shows the description of the complete series of 638 days…”, please review and correct the periods, which are defined differently from that exposed in the Methodology section. “…as well as the pre-vaccination period (March 10, 2020, to December 26, 2020), post-vaccination period (December 27, 2020, to December 31, 2021)”.

7. Correct the period of the first wave in the graph: “10-03-20 to 26-06-20”.

Reply: 

Thank you for noticing these inconsistencies. We have corrected them. 

Reviewer #2:

Comments to the Author

This is a short original article where a temporary disability registry (TD) is assessed as a complimentary system to the standard epidemiological surveillance during the first months of the COVID-19 pandemic in Spain.

The authors should try to improve some of the wording (i.e. Introduction, first paragraph: there is no declaration of a pandemic by WHO: WHO declares a PHEIC; COVID-19 is the disease: SARS-CoV-2 is the etiological agent and cause of the disease).

Reply: 

Thank you very much for pointing this out. We modified the introduction section accordingly:

The COVID-19 was declared by WHO as Public Health Emergency of International Concern (PHEIC) on 30 January 2020 and further characterized as a pandemic on 11 March 2020 [1].

The main concerns related to this paper is the surveillance system that are compared as well as the statistical analysis.

1.- Surveillance system:

The epidemiological surveillance system in Spain was totally overwhelmed in the firsts waves of the pandemic. There were not enough PCR tests and the case definitions was very strictly defined, reducing the number of people tested nearly only to those with severe disease admitted to hospitals as well as Health Care Workers. Is is therefore widely admitted that there was a sizable amount of underestimation of COVID-19 cases.

The TD exceptional surveillance system relies in a personal declaration, usually by phone, where the individual with symptoms and/od that considers him o herself a close contact of a confirmed COVID-case asked for a Temporary Disability leave. There is no any clinical evaluation of the sick leave request, nor any confirmation of the likely diagnosis nor the status of close contact.

Furthermore, Spain institutes a special situation for business and other economic activities called “ERTE” a temporary regulatory employment file, where all or a significant number of workers are entitled to stay at home because of a temporal closure of the economic activity due to the lockdowns. It is likely that many of these workers do not request for a TD leave, albeit they could have been infected and/or close contact of a COVID-19 cases. Both system: being under an ERTE and the TD are incompatible.

In summary, neither the epidemiological surveillance system nor the TD complementary system were both reliable nor accurate systems for assessing the caseload of COVID-19 in Spain over the first waves of the pandemic. As an example, over the period March 10, 2020, to December 31, 2021, the authors state that 4,8 M COVID cases were reported to the epidemiological surveillance system. Our-World-In Data registry indicates more than 6 M COVID-19 cases over the same period.

Several of these limitations (and others as well) are discussed by the authors, but the main conclusion, i.e. that the TD registry could be a useful as a system to complement traditional epidemiological surveillance is weak enough if applied to the COVID-19 pandemic.

Reply:

Thank you for these comments; we consider that they helps us to better understand the context of the pandemic in Spain. We have therefore included in the Discussion some of your points of view considering that enriches the discussion and could improve the understanding of the pandemic data:

The epidemiological surveillance system in Spain was overwhelmed in the first waves of the pandemic. There were not enough PCR tests and the case definitions was very strictly defined, reducing the number of people tested nearly only to those with severe disease admitted to hospitals as well as Health Care Workers. That could lead probable that there was a sizable amount of underestimation of COVID-19 cases. 

Furthermore, Spain institutes a special situation for business called “ERTE”, a temporary regulatory employment file, where all or a significant number of non-essential workers are entitled to stay at home because of a temporal closure of the economic activity due to the lockdown. It is likely that many of these workers do not request for a TD leave, albeit they could have been infected and/or close contact of a COVID-19 cases. In other words, being under an ERTE and the TD are not compatible.

2.- Statistical analysis:

I suggest that the analysis should be reviewed by a statistician.

Reply:

Thank you for this suggestion. The analysis was reviewed by a statistician.

Reviewer #3:

Comments to the Author

The manuscript aimed to explore whether Temporary Disability (TD) could complement the traditional Spanish surveillance system. The authors clearly defined the study's scope, connecting it to current literature. While the methodology and statistical analyses were generally appropriate, some adjustments are needed. More detailed presentation of results is warranted, and the discussion, though robust, brings the benefit from substantiation and comparisons with other studies, along with acknowledgment of significant limitations. This article underscores the potential of complementary systems, traditionally utilized in other countries, to serve as crucial early warning tools for epidemiological surveillance and public health in Spain.

Comments

1- Title suggestion: Temporary Disability (TD) registry as a complementary system to traditional epidemiological surveillance during the COVID-19 pandemic in Spain.

Reply:

Thank you for your suggestion. We agree and have changed the Title.

2- The keywords do not represent the entire manuscript content. Consider incorporating keywords such as Health Surveillance System or Epidemiological Monitoring for a more comprehensive representation.

Reply: 

We have incorporated the suggested keywords.

Temporary disability and Health Surveillance System Sick leave, COVID-19, Spain, Surveillance, Epidemiological Monitoring, 

3- Introduction (Page 3, 4th paragraph) - Insert reference in "In addition, on March 12, another royal decree-law included … and the tourism sector."

Reply: 

We included the suggested reference in the text. 

4- Introduction (Page 3, 7th paragraph) - It would be interesting to explain the delay and difference in the number of notifications between the systems. How do these systems work in Spain?

Reply:

In response to your comments, we have completed the paragraph including this information:

During the pandemic, the main source of information on COVID-19 morbidity was the National Epidemiological Surveillance Network (RENAVE). However, it was observed that the first case reported through RENAVE was recorded eight days after the first TD was granted for infection. In addition, from the beginning of the pandemic until April 21, 2020, the number of TDs due to COVID-19 reported was higher than the number of cases included in RENAVE [9]. They are two independent systems with different objectives. RENAVE depends on the health authorities and seeks epidemiological surveillance. TD system depends on the social security authorities and guarantees the economic income of workers on sick leave from the first day.

More details about the delay and differences between the systems can be found in the Discussion. 

5- Methodology (Page 5, 2th paragraph) - Specify the age range used.

Reply: 

We specified the age range.

6- Methodology (Page 5, 6th paragraph) - Explain the transformations used in IRTD_1 e IRRenave_2.

Reply:

Thank you very much for your suggestion. We have included the following paragraph in response:

The transformations are performed in order to identify the stationarity of the series, in order to identify a series with white noise. The transformation can be set with one lag, two lags or as many lags as necessary, until a time series is found that could be useful for cross-correlation analysis.

7- Methodology (Page 6, 2th paragraph) - It would be interesting to explain better why the 14-day time interval and what was the level of significance adopted.

Reply:

We have included the following commentary in the text in order to specify the use of the 14 days:

The incubation period for the SARS-CoV-2 virus generally ranges from 2 to 14 days. This means that, after being exposed to the virus, most individuals will develop symptoms within that timeframe. That's why an incubation period of up to 14 days is used to encompass the full range in which some individuals may develop symptoms.

8- Results (Pa

---

## [Decision Letter · Decision Letter 1]

27 Feb 2024

PONE-D-23-27681R1Temporary Disability (TD) registry as a complementary system to traditional epidemiological surveillance during the COVID-19 pandemic in Spain a National Study.PLOS ONE

Dear Dr. Culqui L.,

Thank you for submitting your manuscript to PLOS ONE. After careful consideration, we feel that it has merit but does not fully meet PLOS ONE’s publication criteria as it currently stands. Therefore, we invite you to submit a revised version of the manuscript that addresses the points raised during the review process.

We look forward to receiving your revised manuscript.

Kind regards,

Ricardo de Mattos Russo Rafael, Ph.D.

Academic Editor

PLOS ONE

Journal Requirements:

**Additional Editor Comments:**

Please carry out the correction indicated by reviewer 1

Reviewers' comments:

Reviewer's Responses to Questions

**Comments to the Author**

1. If the authors have adequately addressed your comments raised in a previous round of review and you feel that this manuscript is now acceptable for publication, you may indicate that here to bypass the “Comments to the Author” section, enter your conflict of interest statement in the “Confidential to Editor” section, and submit your "Accept" recommendation.

Reviewer #1: All comments have been addressed

Reviewer #2: All comments have been addressed

2. Is the manuscript technically sound, and do the data support the conclusions?

Reviewer #1: Yes

Reviewer #2: (No Response)

3. Has the statistical analysis been performed appropriately and rigorously? 

Reviewer #1: Yes

Reviewer #2: (No Response)

4. Have the authors made all data underlying the findings in their manuscript fully available?

Reviewer #1: Yes

Reviewer #2: (No Response)

5. Is the manuscript presented in an intelligible fashion and written in standard English?

Reviewer #1: Yes

Reviewer #2: (No Response)

6. Review Comments to the Author

Reviewer #1: The paragraph that explains Figure 1, in Results section, remains with incorrect dates in this new version. Please review.

Reviewer #2: (No Response)

7. PLOS authors have the option to publish the peer review history of their article (what does this mean?). If published, this will include your full peer review and any attached files.

Reviewer #1: No

Reviewer #2: No

---

## [Author Response · Author response to Decision Letter 1]

4 Mar 2024

Dear Ms/Mr Editor,

We would like to take this opportunity to thank you for the time dedicated to our article.

We addressed all the comments and suggestions send by the reviewers, and we modified the article accordingly. Below, we are submitting the reviewer´s comments/suggestions and the performed changes:

Review comments to the Author 

Reviewer #1: The paragraph that explains Figure 1, in Results section, remains with incorrect dates in this new version. Please review.

Reviewer #2: (No Response)

Reply: We have reviewed all the information in the paragraph with incorrect dates and have included the correct information in the paper.

---

## [Editor Report · Decision Letter 2]

14 Mar 2024

Temporary Disability (TD) registry as a complementary system to traditional epidemiological surveillance during the COVID-19 pandemic in Spain a National Study.

PONE-D-23-27681R2

Dear Dr. Culqui L.,

We’re pleased to inform you that your manuscript has been judged scientifically suitable for publication and will be formally accepted for publication once it meets all outstanding technical requirements.

Kind regards,

Ricardo de Mattos Russo Rafael, Ph.D.

Academic Editor

PLOS ONE
---

## [Editor Report · Acceptance letter]

26 Apr 2024

PONE-D-23-27681R2 

PLOS ONE

Dear Dr. Culqui L., 

I'm pleased to inform you that your manuscript has been deemed suitable for publication in PLOS ONE. Congratulations! Your manuscript is now being handed over to our production team.

Kind regards, 

on behalf of

Dr. Ricardo de Mattos Russo Rafael 

Academic Editor

PLOS ONE